# Acute pre-exercise hydrogen rich water intake does not improve running performance at maximal aerobic speed in trained track and field runners: A randomized, double-blind, placebo-controlled crossover study

Michal Valenta[1], Michal Botek[1], Jakub Krejčí[1]*, Andrew McKune[2,3], Barbora Sládečková[1], Filip Neuls[1], Robert Bajgar[4,5], Iva Klimešová[1]

1 Department of Natural Sciences in Kinanthropology, Faculty of Physical Culture, Palacký University Olomouc, Olomouc, Czech Republic, 2 Research Institute for Sport and Exercise (UCRISE), University of Canberra, Bruce, Australia, 3 Discipline of Biokinetics, Exercise and Leisure Sciences, School of Health Sciences, University of KwaZulu-Natal, Durban, South Africa, 4 Department of Medical Biophysics, Faculty of Medicine and Dentistry, Palacký University Olomouc, Olomouc, Czech Republic, 5 Institute of Molecular and Translational Medicine, Faculty of Medicine and Dentistry, Palacký University Olomouc, Olomouc, Czech Republic

* jakub.krejci@upol.cz

## Abstract

### Purpose

This study investigated the effects of acute, pre-exercise, hydrogen rich water (HRW) ingestion on running time to exhaustion at maximal aerobic speed in trained track and field runners.

### Methods

Twenty-four, male runners aged 17.5 ± 1.8 years, with body mass index = 21.0 ± 1.3 kg·m$^{-2}$, and maximal oxygen uptake = 55.0 ± 4.6 ml·kg$^{-1}$·min$^{-1}$ (mean ± standard deviation) participated in this randomized, double-blind, placebo-controlled crossover study. All runners ingested 1260 ml of HRW which was divided into four doses and taken at 120 min (420 ml), 60 min (420 ml), 30 min (210 ml), and 10 min (210 ml) prior to exercise. The running protocol consisted of three phases: warm-up performed at 10 km·h$^{-1}$ for 3 min, followed by a transition phase performed at an individually determined speed (10 km·h$^{-1}$ + maximal aerobic speed)/2 for 1 min, and finally the third phase performed at individual maximal aerobic speed until exhaustion. Time to exhaustion, cardiorespiratory variables, and post-exercise blood lactate concentration were measured.

### Results

When running to exhaustion at maximal aerobic speed, compared with placebo, HRW had no significant effects on the following variables: time to exhaustion (217 ± 49 and 227 ± 53 s, $p$ = 0.20), post-exercise blood lactate concentration (9.9 ± 2.2 and 10.1 ± 2.0 mmol·L$^{-1}$,

**Data Availability Statement:** All relevant data are within the article and its Supporting Information files.

**Funding:** This study was supported by the Palacký University Olomouc (URL: www.upol.cz), grant project IGA_FTK_2020_011. The funders had no role in study design, data collection and analysis, decision to publish, or preparation of the manuscript.

**Competing interests:** The authors have declared that no competing interests exist.

$p = 0.42$), maximal heart rate ($186 \pm 9$ and $186 \pm 9$ beats·min$^{-1}$, $p = 0.80$), and oxygen uptake ($53.1 \pm 4.5$ and $52.2 \pm 4.7$ ml·kg$^{-1}$·min$^{-1}$, $p = 0.33$). No variable assessed as a candidate moderator was significantly correlated with time to exhaustion (Spearman's correlation coefficients ranged from $-0.28$ to $0.30$, all $p \geq 0.16$).

## Conclusions

Pre-exercise administration of 1260 ml of HRW showed no ergogenic effect on running performance to exhaustion at maximal aerobic speed in trained track and field runners.

## Introduction

Molecular hydrogen ($H_2$) was initially considered a biologically inactive gas. However, Dole et al. [1] published the first study (mouse model) reporting a significant regression of skin tumors in response to hyperbaric $H_2$ treatment. $H_2$ was shown to have strong selective antioxidative and anti-apoptotic properties, reducing oxidative stress through the scavenging of harmful cytotoxic hydroxyl radicals (OH•) [2]. Recently, Ichihara et al. [3] reported anti-apoptotic, anti-inflammatory, and antioxidative properties of $H_2$. $H_2$ was also considered to be a signaling molecule that contributed to modulation and regulation of gene expression [3]. An anti-fatigue effect of $H_2$ in response to pre-exercise intake of hydrogen rich water (HRW) was shown across different modes of exercise, specifically, repeated isokinetic knee extensions [4], intermittent cycling sprints [5], repeated running sprints [6], anaerobic performance [7], and strength-endurance drills [8]. Research also showed that an anti-fatigue effect of pre-exercise HRW intake seemed to be dependent on the current performance status of athletes [7, 9]. Further, studies demonstrated that pre-exercise $H_2$ exposure led to a lower rate of perceived exertion (RPE), enhanced ventilation efficiency, reduction in blood lactate concentration [10], and stimulated prefrontal cortex activity [11], particularly during higher exercise intensities. In addition, acute HRW intake resulted in a post-exercise lactate lowering effect [4, 12] and lower delayed onset of muscle soreness after strength exercise [8, 13]. However, pre-exercise HRW consumption did not positively affect submaximal running performance, physiological responses, or time to exhaustion, in response to a maximal incremental running test [14]. In addition, no significant difference was reported between HRW and control groups for time to exhaustion in an incremental cycling test in a heated environment [15], and there was no ergogenic effect of $H_2$ during an incremental maximal test in either amateur or professional cyclists [7]. Despite the inconsistent exercise ergogenic effects of $H_2$, recent *in vitro* studies have reported mitochondrial effects of $H_2$. $H_2$ was shown to increase mitochondrial oxygen consumption rate, stimulate mitochondrial Q cycle and enhance oxidative adenosine triphosphate production [16, 17].

Based on the potential aerobic energy system benefits of $H_2$, this study was designed to investigate whether acute HRW supplementation improved running performance at maximal aerobic speed. From a practical application standpoint, maximal aerobic speed is closely related to the running velocity that can be sustained by elite runners over 3000 m [18]. Therefore, the primary aim of this study was to assess the effect of acute pre-exercise HRW intake on time to exhaustion when running at maximal aerobic speed in trained track and field runners. We hypothesized that the anti-fatigue effect of pre-exercise HRW ingestion [4, 6, 8, 10] would improve running performance at maximal aerobic speed with an increase in time to exhaustion.

## Methods

### Participants

The primary inclusion criterion was a personal best time in the 1500 m run of under 4:33.0 for adult participants and 5:16.0 for participants younger than 18 years. This time should have been achieved in regular competition no longer than one year before the experiment. Seventy-two potential participants were contacted, with thirty indicating that they were interested in participating in the study. Two participants withdrew before the first session and four did not complete the experiment due to medical complications or technical problems during testing (Fig 1). Twenty-four, young, male, trained track and field runners (I. and II. Czech national track and field league competitors) successfully finished this study (Table 1).

Prior to testing, all participants were informed about the aim of the study and the testing procedures. All participants were asked to complete health questionnaire to demonstrate that they were free of any health problems. The research was conducted in accordance with the Declaration of Helsinki and was approved by the Ethics Committee of the Faculty of Physical

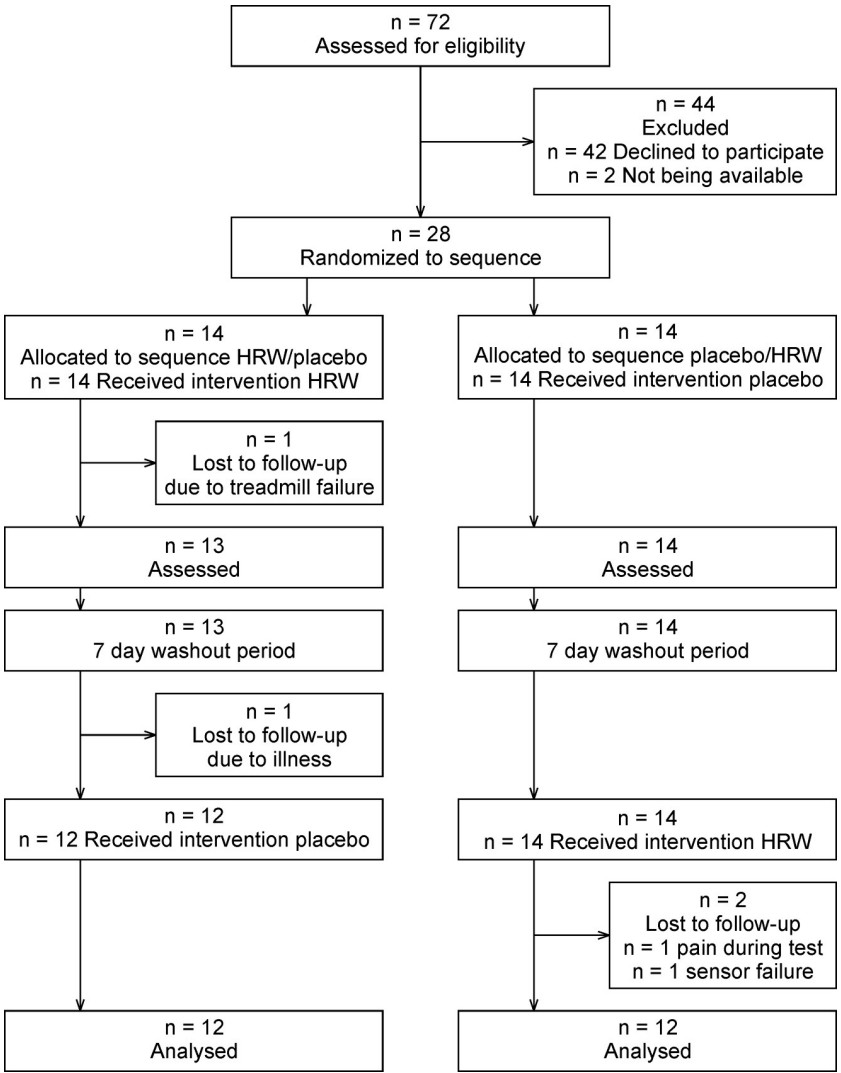

**Fig 1. CONSORT flow diagram.** HRW = hydrogen rich water.

Table 1. Characteristics of the runners (n = 24).

| Variable | p | Mean ± SD | Median (Q1, Q3) |
|---|---|---|---|
| Age (years) | 0.014 | 17.5 ± 1.8 | 17.5 (16.0, 18.5) |
| Body mass (kg) | 0.43 | 69.1 ± 5.9 | 69.5 (64.4, 73.7) |
| Body height (cm) | 0.018 | 181.5 ± 5.5 | 183 (177, 186) |
| BMI (kg·m⁻²) | 0.12 | 21.0 ± 1.3 | 20.6 (20.0, 21.7) |
| Body fat (%) | 0.091 | 10.1 ± 4.6 | 9.1 (6.3, 14.8) |
| $VO_2max$ (ml·kg⁻¹·min⁻¹) | 0.30 | 55.0 ± 4.6 | 54.3 (51.0, 57.4) |
| Pmax (W·kg⁻¹) | 0.14 | 6.05 ± 0.55 | 6.06 (5.73, 6.22) |
| ANT (beats·min⁻¹) | 0.78 | 180 ± 9 | 179 (174, 184) |
| HRmax (beats·min⁻¹) | 0.61 | 196 ± 9 | 196 (190, 201) |
| MAS (km·h⁻¹) | 0.003 | 18.3 ± 1.5 | 18.0 (17.5, 18.5) |

$p$ = statistical significance (Shapiro-Wilk test); SD = standard deviation; Q1 = the first quartile; Q3 = the third quartile; BMI = body mass index; $VO_2max$ = maximal oxygen consumption; Pmax = maximal power output; ANT = anaerobic threshold; HRmax = maximal heart rate; MAS = maximal aerobic speed.

Culture, Palacký University Olomouc (reference number 9/2020). Participation in this research was voluntary and all participants signed informed consent. If participants were <18 years of age, written parental consent was obtained.

## Experimental design

The study had a randomized, double-blind, placebo-controlled crossover design. The participants attended three laboratory sessions and one outdoor training session (Fig 2). The aim of the first laboratory session was to provide information related to the experiment, obtain anthropological data, ensure familiarization with equipment, and determine individual maximal aerobic speed. For three days prior to the first laboratory testing, participants did not participate in any strenuous activity.

The second and the third laboratory sessions included the running protocol in which the effect of HRW supplementation was examined. The second laboratory session took place three days after the first laboratory session. The wash out period between the second and the third laboratory session was 7 days. All participants performed one outdoor training session 3 days before the third laboratory session. This training session was included to maintain two identical microcycles. Exercise load and intensity were individually set and corresponded with the maximal aerobic speed testing performed during the first laboratory session.

All participants were randomly divided into two groups, HRW/placebo or placebo/HRW sequences. For the randomization process, three red and three blue paper strips were placed in a sachet. Each participant was asked to draw one strip whilst being blinded. When the sachet was empty, it was refilled with three red and three blue strips and the the procedure was continued until all participants were randomized. According to Kang et al. [19], this procedure can be described as block randomization, which prevents an unequal number of participants in two groups. Participants who pulled out the red strips received water packages with batch number A in the second session, and batch number B in the third session. Participants who pulled out the blue strips received water packages with batch number B in the second session and batch number A in the third session. After the statistical analysis was finished, the manufacturer of the HRW and placebo provided the researchers with the details regarding which batch numbers were HRW and placebo.

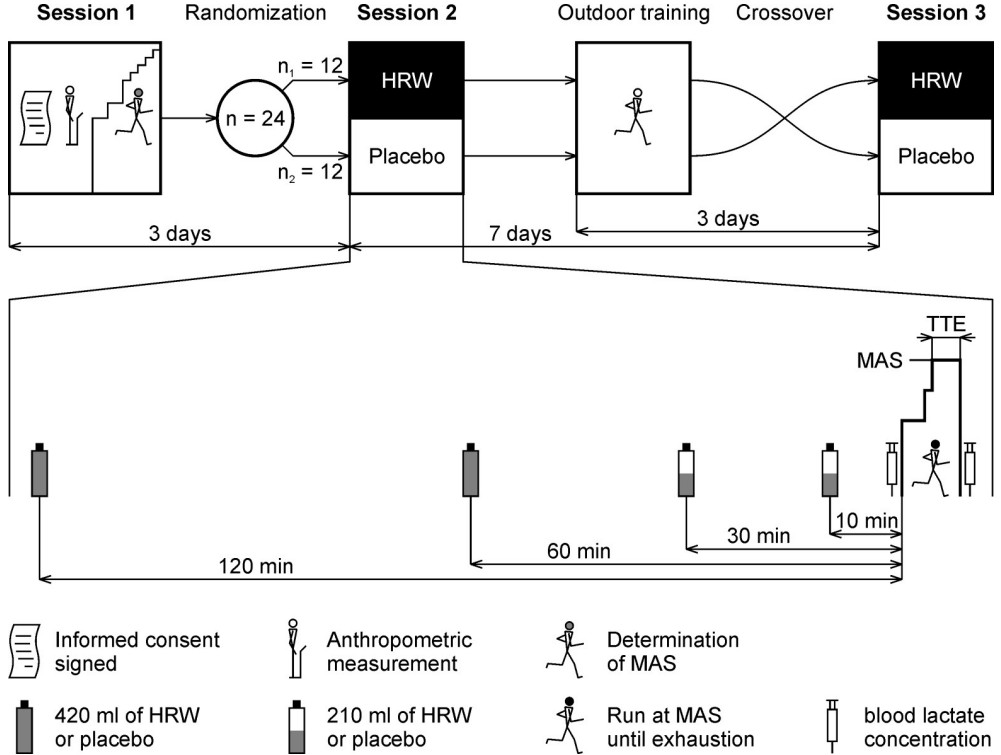

**Fig 2. Overview of the study protocol and labelling of sessions.** HRW = hydrogen rich water; TTE = time to exhaustion; MAS = maximal aerobic speed.

All laboratory sessions were performed under standardized conditions. Room temperature was maintained at 22–24°C, with relative air humidity maintained between 40 and 60%. Every participant was allocated their own testing time which was constant for all sessions to prevent possible circadian influence. Participants were instructed to avoid drinking coffee, tea or other substances in the two hours before testing as these substances may potentially affect selected physiological or perceptual responses. Furthermore, participants were asked to avoid drinking alcohol 48 h before all laboratory sessions, and to avoid (other than the prescribed maximal aerobic speed testing or outdoor training session) vigorous physical activity three days prior to the first laboratory session and between the following sessions.

## Anthropometric measurement

Body height was measured to the nearest 1 cm using a standardized stadiometer. Body mass (to nearest 0.1 kg) and percentage body fat (bioimpedance analysis) were determined using the Tanita BC-418 MA (Tanita, Tokyo, Japan).

## Determination of maximal aerobic speed

Individual maximal aerobic speed was determined using a stepwise, incremental protocol performed on a treadmill Lode Valiant Special (Lode, Groningen, Netherlands) while gas exchange, ventilatory characteristics (Ergostik, Geratherm Respiratory, Bad Kissingen, Germany) and heart rate (Polar, Kempele, Finland) were recorded [20]. There was an initial warm up at 10 km·h$^{-1}$ (2 min) and 12 km·h$^{-1}$ (2 min), which was followed by an individual number of 1 min incremental steps starting at a speed of 15 km·h$^{-1}$, with each step increasing in speed

by 1 km·h$^{-1}$. The test was performed until voluntary exhaustion. The criteria for attaining VO$_2$max was defined as reaching one of the following criteria: a) respiratory exchange ratio of >1.11 [21]; b) VO$_2$ plateau defined as no increase in VO$_2$ in response to an increase in work rate [22]. VO$_2$max was considered the highest VO$_2$ value in the final 30 s of the test [23]. Maximal aerobic speed was defined as the minimal running speed that elicited VO$_2$ equal to VO$_2$max, with the participant able to finish the 1 min step at this speed.

## Running protocol

The running protocol was divided into three steps. The first 3 min warm up step (10 km·h$^{-1}$) was followed by a 1 min step at an individually set speed (10 km·h$^{-1}$ + maximal aerobic speed)/2 used for smooth transition to the third step. The third step was performed at individual maximal aerobic speed until exhaustion. The time to exhaustion was measured to the nearest 1 s. Both tests in the second and third sessions were performed by the same tester, who was instructed to avoid verbal communication with the participant during the testing. Ventilation and gas exchange were recorded breath by breath. Heart rate was recorded continuously. The average values of the last 30 s were calculated for statistical analysis. Immediately after finishing the test, blood samples were collected to determine the blood lactate concentration using a Lactate Scout + (EKF Diagnostic, Cardiff, United Kingdom). The blood samples were collected, and instrument accuracy was checked, according to the manufacturer guidelines.

## HRW and placebo chemical composition, administration strategy

A total volume of 1260 ml of HRW (Aquastamina-R HRW, Nutristamina, Ostrava, Czech Republic) or placebo (Aquastamina-R placebo, Nutristamina, Ostrava, Czech Republic) was administered in four doses, specifically 420 ml of HRW/placebo was applied 120 min and 60 min before exercise, and 210 ml of HRW/placebo was applied 30 min and 10 min before exercise. This HRW hydration protocol included a one-week washout period similarly to previous HRW studies [4, 9, 10]. According to manufacturer information, HRW was produced by infusing H$_2$ under high pressure directly into the water. Both drinks were served in visually identical plastic-aluminum packages. Participants could not distinguish between HRW and placebo because H$_2$ is colorless, odorless, and tasteless [24]. The chemical properties of both HRW and placebo (Table 2) were determined using the pH/ORP/Temperature-meter (AD14, Adwa Instruments, Szeged, Hungary). The dissolved H$_2$ concentration was determined using H2Blue reagent (H2 Sciences, Henderson, NV, USA) according to the manufacturer instructions.

## Statistical analysis

The normality of data was verified using the Shapiro-Wilk test. Data are presented as arithmetic mean ± standard deviation or median (the first quartile, the third quartile). To obtain

**Table 2. Physico-chemical properties of hydrogen rich water and placebo water.**

| Property | HRW | Placebo |
|---|---|---|
| pH | 7.8 | 7.6 |
| ORP (mV) | -659 | +172 |
| Temperature (˚C) | 22 | 22 |
| H$_2$ concentration (ppm) | 0.9 | 0.0 |

HRW = hydrogen rich water; ORP = oxidation reduction potential.

percentage changes, the data were logarithmically transformed, statistically processed, back transformed, and expressed as percentages. These transformations were performed using a specialized spreadsheet [25]. This spreadsheet was also used to estimate the reliability of time to exhaustion expressed as a coefficient of variation. The normality of logarithmically transformed data was also verified using the Shapiro-Wilk test. The effect of HRW compared to placebo was evaluated using a paired two-tailed t-test. The effect size was evaluated using Cohen's $d$ according to the formula $d = m_\Delta / SD_{Pla}$, where $m_\Delta$ is the mean of difference scores between HRW and placebo and $SD_{Pla}$ is the standard deviation calculated from the placebo values. The following thresholds for interpreting the magnitude of the effect size were used [26]: 0.00–0.19 trivial, 0.20–0.59 small, 0.60–1.19 moderate, ≥1.20 large.

In order to examine the individual responses of time to exhaustion, the smallest worthwhile change (SWC) was determined and the frequencies of positive responders (Δ ≥ SWC), non-responders (SWC > Δ > −SWC), and negative responders (Δ ≤ −SWC) were calculated by comparing the individual difference score (Δ = HRW − Placebo) against the three intervals defined by SWC. The SWC for competitive runners is ~0.3% of performance time [27]. Hinckson and Hopkins [28] reported that a 1% change in time trial performance leads to a 10–20% change in time to exhaustion. Therefore, the SWC for time to exhaustion was set at 0.3% × 10 = 3%. The significance of the odds ratio of positive/negative responders was evaluated using a chi-square test.

Variables considered as candidate moderators modifying the effect of HRW on time to exhaustion were examined by correlation analysis using Spearman's correlation coefficient. For all statistical tests, $p < 0.05$ was considered statistically significant. Statistical analyses were performed using MATLAB version R2020a (MathWorks, Natick, MA, USA) and specialized spreadsheet [25]. Based on Botek et al. [10] we expected the effect size in this study to be at least moderate ($d ≥ 0.6$). A priori power analysis considering a paired two-tailed t-test was performed using G*Power version 3.1.9.7 [29] with parameters $d = 0.6$, $\alpha = 0.05$, and $\beta = 0.20$. The desired sample size resulted in 24 participants.

## Results

Raw data are available in S1–S3 Tables. The anthropometric and performance characteristics of participants are presented in Table 1. Each participant received 567 μmol of $H_2$ dissolved in 1260 ml of HRW during the experimental running protocol. The dose relative to body mass was 8.26 ± 0.71 μmol·kg$^{-1}$ expressed as mean ± standard deviation or 8.17 (7.70, 8.80) μmol·kg$^{-1}$ expressed as median (first and third quartiles).

After logarithmic transformation, all variables listed in Table 3 had normal distributions (all $p ≥ 0.073$, Shapiro-Wilk test) except BF ($p = 0.001$). The departure from normality was considered small after visual inspection of the quantile-quantile plot and the BF variable was also processed using a t-test because it is considered robust for such departure from normality. The coefficient of variation for time to exhaustion estimated from this crossover study was 13% with a 95% confidence interval of 10 to 18%. The effects of HRW on performance and physiological variables are presented in Table 3. No statistically significant effects of HRW were found (all $p ≥ 0.20$, paired t-test). Absolute Cohen's d values ranged from 0.01 to 0.19, indicating trivial effects. Analysis of individual responses of time to exhaustion (Fig 3) revealed that 12 runners responded positively to HRW, 3 runners did not respond, and 9 runners responded negatively. The odds ratio of positive/negative responders (12/9) was not significant ($p = 0.51$, chi-square test). Therefore, the hypothesis that acute HRW supplementation has an ergogenic effect in trained runners running at maximal aerobic speed was rejected.

**Table 3. Effect of hydrogen rich water compared to placebo on performance and physiological variables.**

| Variable | HRW | Placebo | $d$ | $\Delta$ | $p$ |
|---|---|---|---|---|---|
| | Mean ± SD | Mean ± SD | | (95% CI) | |
| TTE (s) | 217 ± 49 | 227 ± 53 | −0.19 | −4.4 (−11.0 to 2.6) | 0.20 |
| DTE (m) | 1096 ± 254 | 1144 ± 262 | −0.18 | −4.4 (−11.0 to 2.6) | 0.20 |
| La pre (mmol·L⁻¹) | 1.6 ± 0.4 | 1.6 ± 0.4 | 0.01 | 0.4 (−9.0 to 10.8) | 0.93 |
| La post (mmol·L⁻¹) | 9.9 ± 2.2 | 10.1 ± 2.0 | −0.11 | −2.8 (−9.7 to 4.6) | 0.42 |
| HR (beats·min⁻¹) | 186 ± 9 | 186 ± 9 | −0.02 | −0.1 (−0.7 to 0.6) | 0.80 |
| BF (breaths·min⁻¹) | 59 ± 12 | 60 ± 13 | −0.03 | −0.4 (−3.0 to 2.3) | 0.77 |
| VE (ml·kg⁻¹·min⁻¹) | 2099 ± 222 | 2083 ± 214 | 0.08 | 0.8 (−2.4 to 4.0) | 0.63 |
| VO₂ (ml·kg⁻¹·min⁻¹) | 53.1 ± 4.5 | 52.2 ± 4.7 | 0.18 | 1.7 (−1.8 to 5.2) | 0.33 |
| VCO₂ (ml·kg⁻¹·min⁻¹) | 61.4 ± 4.6 | 60.6 ± 5.2 | 0.15 | 1.4 (−1.5 to 4.4) | 0.34 |
| VE/VO₂ | 39.7 ± 4.1 | 40.0 ± 3.8 | −0.09 | −0.9 (−3.0 to 1.2) | 0.40 |
| RQ | 1.159 ± 0.054 | 1.163 ± 0.066 | −0.06 | −0.3 (−2.2 to 1.7) | 0.76 |

HRW = hydrogen rich water; SD = standard deviation; $d$ = Cohen's d effect size; $\Delta$ = effect of HRW compared to placebo expressed as a percentage; CI = confidence interval; $p$ = statistical significance (paired t-test); TTE = time to exhaustion; DTE = distance to exhaustion; La pre = pre-exercise blood lactate concentration; La post = post-exercise blood lactate concentration; HR = heart rate; BF = breathing frequency; VE = ventilation; VO₂ = oxygen uptake; VCO₂ = carbon dioxide release; VE/VO₂ = ventilatory equivalent for oxygen; RQ = respiratory quotient.

Two candidate moderators, namely age and maximal aerobic speed (Table 1), had non-normal distributions, so correlation analysis was performed using the non-parametric Spearman's correlation coefficient. Values of Spearman's correlation coefficient ranged from −0.28 to 0.30 and these values were not statistically different from zero (all $p \geq 0.16$, Table 4). Specifically, runner performance level expressed as pooled time to exhaustion (average of HRW and placebo values) could not be considered a suitable moderator of the effect of HRW on time to exhaustion ($r = 0.12$, $p = 0.56$, Fig 4). Thus, no suitable moderator of the effect of HRW was found in this study.

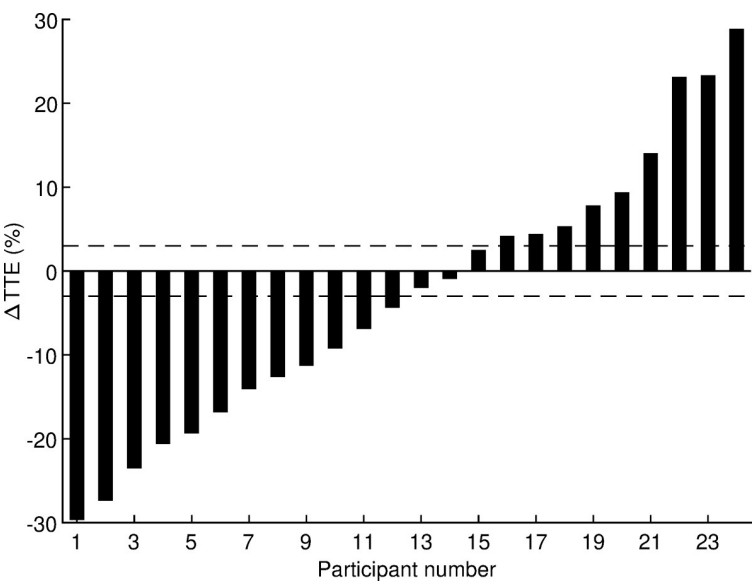

**Fig 3. Individual percentage change in time to exhaustion.** $\Delta$ = difference between hydrogen rich water and placebo; TTE = time to exhaustion. The horizontal lines represent the smallest worthwhile change.

**Table 4. Correlation analysis between effect of hydrogen rich water on time to exhaustion and various variables.**

| Variable | $r_S$ | $p$ |
| --- | --- | --- |
| $(\text{TTE}_{\text{HRW}} + \text{TTE}_{\text{Pla}})/2$ | 0.12 | 0.56 |
| Age | 0.03 | 0.90 |
| Body mass | 0.03 | 0.88 |
| BMI | −0.10 | 0.64 |
| Body fat | −0.28 | 0.19 |
| VO$_2$max | 0.23 | 0.28 |
| Pmax | 0.29 | 0.17 |
| ANT | 0.09 | 0.67 |
| HRmax | 0.07 | 0.75 |
| MAS | 0.30 | 0.16 |
| H$_2$ dose | −0.03 | 0.88 |

$r_S$ = Spearman's correlation coefficient; $p$ = statistical significance of correlation coefficient; TTE = time to exhaustion; HRW = hydrogen rich water; Pla = placebo; BMI = body mass index; VO$_2$max = maximal oxygen consumption; Pmax = maximal power output; ANT = anaerobic threshold; HRmax = maximal heart rate; MAS = maximal aerobic speed;

## Discussion

The primary aim of this study was to assess the influence of a dose of 1260 ml HRW on running performance as indicated by time to exhaustion at maximal aerobic speed in trained track and field runners. Based on previous studies that have demonstrated an anti-fatigue effect of HRW intake before exercise [4–6, 8, 10, 12], we hypothesized that HRW intake prior to exercise would improve maximal aerobic speed performance, specifically running time to exhaustion. Contrary to our hypothesis, we observed no significant effect of HRW in time to

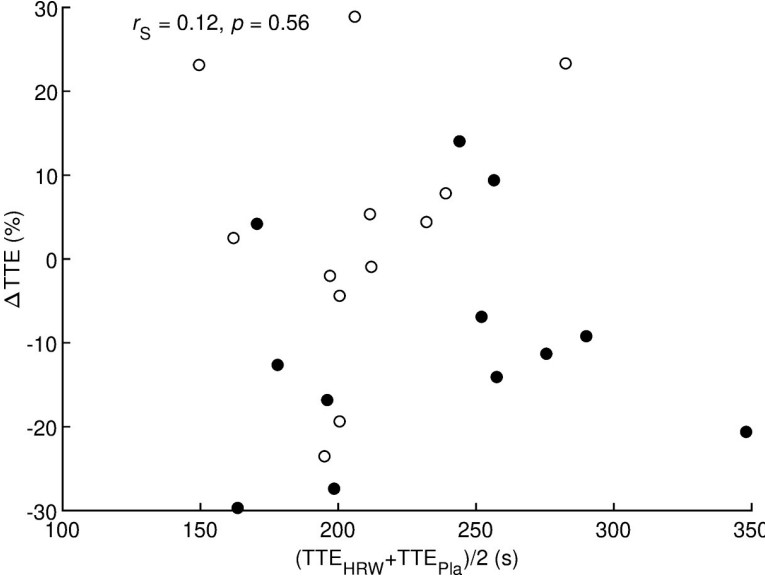

**Fig 4. Correlation analysis between effect of hydrogen rich water on time to exhaustion and pooled time to exhaustion.** $r_S$ = Spearman's correlation coefficient; $p$ = statistical significance of correlation coefficient; Δ = difference between hydrogen rich water and placebo; TTE = time to exhaustion; HRW = hydrogen rich water; Pla = placebo. Filled and open circles indicate runners who received HRW and placebo, respectively, in the first test.

exhaustion, as a valuable predictor of 1500 m running performance [30], or any other physiological variable during running at maximal aerobic speed.

It is accepted that running to exhaustion at maximal aerobic speed represents a very high exercise intensity, where a substantial anaerobic glycolytic contribution towards adenosine triphosphate resynthesis is involved, and consequently, there is an onset of blood lactate accumulation [31]. In the current study, trained track and field runners ran at a determined speed of $18.3 \pm 1.5$ km·h$^{-1}$ for a duration of $217 \pm 49$ s after ingesting HRW, and for $227 \pm 53$ s, after ingesting a placebo. This running velocity corresponds with a time to exhaustion ranging from 2.5 to 10 min for different kinds of activities [32], while it is slower compared with elite and sub-elite middle- and long-distance runners [32, 33]. From a practical standpoint, if maximal aerobic speed and time to exhaustion describes 95% of variance in the average race velocity over 1500 m [30], we suggest that the enhancement for middle-distance performance by acute supplementation of 1260 ml HRW would also be ineffective. In addition, the post-exercise blood lactate concentration was $9.9 \pm 2.2$ mmol·L$^{-1}$ and $10.1 \pm 2.0$ mmol·L$^{-1}$ for HRW and placebo, respectively. These values indirectly indicated exercise-induced metabolic acidosis [34] and potential peripheral fatigue development [35]. Besides the increasing muscle acidosis, an excessive production of reactive oxygen species during exhaustive running [36, 37] may have also contributed to the deterioration of muscle performance, as previously demonstrated [38].

The results show that the expected antifatigue effect of $H_2$ was not detected when a dose of 1260 ml HRW ($H_2 = 0.9$ ppm) was ingested prior to the run at maximal aerobic speed with a duration of up to ~230 s. We assume that the anticipated antifatigue effects of $H_2$ was likely prevented by the increasing muscle acidosis, together with excessive oxidative stress. This result is consistent with some previous studies that examined the acute ingestion of HRW before exercise. For instance, an acute intake dose of 290 ml HRW ($H_2 = 1.0$ ppm) before testing and 290 ml HRW during a 10 min rest period between submaximal and maximal sections of an experimental protocol did not affect cardiorespiratory and metabolic variables during an incremental submaximal running test (34–91% $VO_2$max) or the subsequent ~619 s running to exhaustion during incremental exercise [14]. These authors concluded that two doses of 290 ml of HRW before incremental running to exhaustion was not sufficiently ergogenic in endurance-trained athletes [14]. Similarly to our findings, Ito et al. [15] reported that an applied dose of 2.0 ml·kg$^{-1}$ of either HRW or placebo, ingested every 15 min within 60 min of cycling at 65% of $VO_2$max, followed by an incremental cycling test to exhaustion, did not improve performance in well trained triathletes. In this context, Nogueira et al. [39] reported no significant changes in running time to exhaustion in rats after inhaling either 2% $H_2$ or $H_2$ free air during acute, exhaustive physical exercise. Interestingly, despite no ergogenic effect of $H_2$ inhalation on running performance, post-exercise biochemical analysis revealed important findings, where $H_2$ inhalation was associated with effective downregulation of muscle damage, reducing oxidative stress, inflammation, and apoptosis after exhaustive acute exercise in the rats that were unaccustomed to this level of exercise [39].

From methodological standpoint, the question arises whether our results may be tainted by small sample size or poor reliability of measurement. Before starting the experimental part, we performed a power analysis based on the results of Botek et al. [10]. From this analysis the desired sample size was calculated to be 24 participants, which was honored in the experiment. Previous studies [4, 5, 10] used even smaller sample sizes [8, 10 and 12, respectively] and found that HRW improved responses to exercise, including a reduction in lactate concentration and fatigue. Therefore, the sample size in this study is unlikely to be too small. The reliability of the time to exhaustion in this study was 13%, expressed as a coefficient of variation. Although this value was obtained from an intervention study and not a reliability study, it is comparable to values in reliability studies: 11% [32], 13% [40], indicating a good level of methodology and

standardization of measurement in our laboratory. The coefficient of variation of time to exhaustion in constant-power tests is known to be highest in physical performance tests [28]. However, Hopkins et al. [41], using the relationship between exercise duration and power output, showed that the reliability of the equivalent mean power calculated from the constant-power test (0.6%) is even better than from the constant-work test (1.0%). We therefore consider time to exhaustion as a suitable index for tracking changes in performance.

Contrary to our current findings, there are four studies that have reported an anti-fatigue or performance enhancing effect of HRW intake prior to exercise. Specifically, an antifatigue effect of HRW (2 L per day for 2 weeks pre-exercise, dissolved $H_2$ = 0.15 to 0.45 ppm, and pH = 9.8.) during intermittent cycling was demonstrated by Da Ponte et al. [5], who reported a 7.4% attenuation in the decline of peak power output from the 6[th] to the 9[th] of 10 sprints. In addition, in soccer players, Aoki et al. [4] reported an attenuated decrease (3.7%) in peak torque for 20 isokinetic knee extensions, following 30 min of cycle ergometry at an intensity of 75% $VO_2$, when 1.5 L of HRW ($H_2$ = 1.84–2.04 ppm) was ingested within 8 hours pre-exercise. Positive effects of acute, intermittent ingestion of HRW on resistance training was recently demonstrated by Botek et al. [8]. Ingestion of 1260 ml HRW ($H_2$ = 0.9 ppm) pre-exercise resulted in lower blood lactate concentration, improve muscle function, and alleviated muscle pain perception in physical active males.

Importantly, it is very difficult to accurately compare our findings with the previously mentioned studies due to the variable methodology used, including exercise protocols (mode, intensity, duration), dose of HRW and its chemical properties (concentration of dissolved $H_2$, and its pH), and the training status of involved participants. These variables all impact the body response to a particular mode of exercise. For instance, Botek et al. [9] recently showed that pre-race hydration with 1680 ml of HRW improved endurance running performance by 1.3% in the slowest runners, whilst the effect of HRW on race performance in the fastest runners was unclear (deterioration by 0.8%), and concluded that the magnitude of anti-fatigue effect of $H_2$ depended on individual adaptation level. The results suggested that faster athletes seem to be less sensitive to acute $H_2$ supplementation compared with slower athletes who exhibit higher benefits from acute $H_2$ intake. In contrast, Timón et al. [7] recently reported an improved anaerobic performance in a group of trained cyclists after one week of HRW administration ($H_2$ = 1.9 ppm, dose of ~2 L per day) compared with no improvements in a group of amateur cyclists. In the current study, we did not find a suitable moderator of individual responses of time to exhaustion, which may have been due to the homogeneity of running performance in our cohort of runners. A future study involving athletes with more heterogenous performance is needed to assess this phenomenon.

Considering our results and previous research, we suggest that acute $H_2$ intake before exercise does not always provide ergogenic effects. Based on current results, we could not recommend acute pre-exercise HRW intake as an ergogenic supplement to improve time to exhaustion at maximal aerobic speed. There are some limitations and issues regarding HRW application in this study. 1) oxidative stress variables were not assessed. It appears that this information may be helpful for a deeper understanding of how $H_2$ may affect ROS production and performance responses. 2) the dosage of $H_2$ was constant for all participants for logistical reasons and was not adjusted to body mass. 3) Several variables were compared, and no technique was used to control for Type 1 statistical error.

## Conclusions

This study found that a dose of 1260 ml HRW ingested prior to an exhaustive run had no significant effect on the running performance at maximal aerobic speed in a cohort of the

national level track and field runners. Acute ingestion of HRW as a hydration strategy prior to exercise in trained middle-distance runners is not recommended to improve performance.

## Supporting information

**S1 Table. Raw data for Table 1.**
(XLSX)

**S2 Table. Raw data for Table 3, part 1.**
(XLSX)

**S3 Table. Raw data for Table 3, part 2.**
(XLSX)

## Author Contributions

**Conceptualization:** Michal Valenta, Michal Botek.

**Data curation:** Michal Valenta, Michal Botek, Jakub Krejčí.

**Formal analysis:** Jakub Krejčí, Andrew McKune.

**Funding acquisition:** Michal Valenta.

**Investigation:** Michal Valenta, Michal Botek, Jakub Krejčí, Barbora Sládečková, Filip Neuls, Robert Bajgar, Iva Klimešová.

**Methodology:** Michal Valenta, Michal Botek, Jakub Krejčí.

**Project administration:** Michal Valenta.

**Supervision:** Michal Botek.

**Visualization:** Jakub Krejčí.

**Writing – original draft:** Michal Valenta, Michal Botek.

**Writing – review & editing:** Jakub Krejčí, Andrew McKune, Barbora Sládečková, Filip Neuls, Robert Bajgar, Iva Klimešová.

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
