## [Decision Letter · Decision Letter 0]

11 Aug 2022

PONE-D-22-15394Acute pre-exercise hydrogen rich water intake does not improve running performance at maximal aerobic speed in trained track and field runnersPLOS ONE

Dear Dr. Krejčí,

Thank you for submitting your manuscript to PLOS ONE. After careful consideration, we feel that it has merit but does not fully meet PLOS ONE’s publication criteria as it currently stands. Therefore, we invite you to submit a revised version of the manuscript that addresses the points raised during the review process.

We look forward to receiving your revised manuscript.

Kind regards,

Walid Kamal Abdelbasset, Ph.D.

Academic Editor

PLOS ONE

Journal Requirements:

Reviewers' comments:

Reviewer's Responses to Questions

**Comments to the Author**

1. Is the manuscript technically sound, and do the data support the conclusions?

Reviewer #1: Yes

Reviewer #2: Yes

2. Has the statistical analysis been performed appropriately and rigorously? 

Reviewer #1: Yes

Reviewer #2: Yes

3. Have the authors made all data underlying the findings in their manuscript fully available?

Reviewer #1: Yes

Reviewer #2: No

4. Is the manuscript presented in an intelligible fashion and written in standard English?

Reviewer #1: Yes

Reviewer #2: Yes

5. Review Comments to the Author

Reviewer #1: A randomized placebo-controlled crossover research study aimed to investigate the effects of ingesting hydrogen rich water (HRW) on running time in track and field runners (n=24). Time to exhaustion, post-exercise blood lactate concentration, maximal heart rate, and oxygen uptake were not significantly different between placebo and HRW.

Minor revisions:

1- Abstract: Define r.

2- Table 1: Indicate if the distribution of the variables were checked for normality. For nonparametric data, summarize using median, first and third quartiles.

3- Line 229: Within the text, indicate the statistical method used to estimate the correlations. Since the p-value tests the null hypothesis that the correlation equals zero, it may be more precise to state that the correlations were not statistically different than zero.

Reviewer #2: Title: Acute pre-exercise hydrogen rich water intake does not improve running performance at maximal aerobic speed in trained track and field runners

GENERAL COMMENTS

This study investigated the effects of acute, pre-exercise, hydrogen-rich water (HRW) ingestion on running time to exhaustion at maximal aerobic speed in trained track and field runners. I have read with attention the previous reviewers’ comments. I feel that the study is interesting, but however, I found that there are several challenges in carry out other studies, due to a prori no results may not directly to indicate that exercise hydrogen-rich water intake does not improve running performance, it was evaluated at a particular condition (maximal aerobic speed), but not at anaerobic speed, and with a “time” as a performance marker, but not at other markers (efficiency of running, glycogen content, fat, CHO etc), and with specific sample characteristics (young runners). Additionally, the study protocol and design it is high-quality, and easy to replicate, then I like to recommend the study for future other studies that can help us to carry out other more complex studies.

ABSTRACT

It is well design

INTRODUCTION

-I like to suggest to the authors to include a little more information regarding that pre-exercise H2 ingestion have been reported not at all exercise modalities, intensities, volumes, nor including other outocmes more thatn the “time” in a performance, and population (runners trained).

Methods

I found the methods reported in a very good quality

The study protocol figure is very understandable

Statistical analyses are very clear.

RESULTS

There are very clear described.

DISCUSSION

-I like to suggest to the authors to include some future potential other study conditions as I have stated previously (i.e., other populations), where this previous study give us only a high-quality, but very specific condition in which the hydrogen rich water (HRW) ingestion do not add new advantages.

Tables

Are very clear

Figures

Are very clear

References

Are very up-dated.

6. PLOS authors have the option to publish the peer review history of their article (what does this mean?). If published, this will include your full peer review and any attached files.

Reviewer #1: No

Reviewer #2: **Yes: **Cristian Alvarez

---

## [Author Response · Author response to Decision Letter 0]

19 Sep 2022

Reviewer #1

A randomized placebo-controlled crossover research study aimed to investigate the effects of ingesting hydrogen rich water (HRW) on running time in track and field runners (n=24). Time to exhaustion, post-exercise blood lactate concentration, maximal heart rate, and oxygen uptake were not significantly different between placebo and HRW.

1– Abstract: Define r.

Response: Thank you for your comment. The abbreviation was replaced with the full title and the sentence now reads “Spearman’s correlation coefficients ranged from −0.28 to 0.30, all p ≥ 0.16”.

2– Table 1: Indicate if the distribution of the variables were checked for normality. For nonparametric data, summarize using median, first and third quartiles.

Response: Thank you for this suggestion. It is our mistake that we neglected to check the normality assumption. We added the results of the Shapiro-Wilk test. Three variables were significantly different from the normal distribution. We therefore added median, first and third quartile values to Table 1. In the correlation analysis we switched to a non-parametric alternative – Spearman’s correlation coefficient.

3– Line 229: Within the text, indicate the statistical method used to estimate the correlations. Since the p-value tests the null hypothesis that the correlation equals zero, it may be more precise to state that the correlations were not statistically different than zero.

Response: We re-read the manuscript and found that there was no clear distinction between the analysis of individual responses and correlation analysis. We rearranged the text and hope this helped to make it clearer. Spearman’s correlation coefficient as the statistical method used was added to the text. Thank you for your comment regarding the null hypothesis of zero correlation. We rewrote the sentence. Please see the Results section in the manuscript.

Reviewer #2

GENERAL COMMENTS

This study investigated the effects of acute, pre-exercise, hydrogen-rich water (HRW) ingestion on running time to exhaustion at maximal aerobic speed in trained track and field runners. I have read with attention the previous reviewers’ comments. I feel that the study is interesting, but however, I found that there are several challenges in carry out other studies, due to a prori no results may not directly to indicate that exercise hydrogen-rich water intake does not improve running performance, it was evaluated at a particular condition (maximal aerobic speed), but not at anaerobic speed, and with a “time” as a performance marker, but not at other markers (efficiency of running, glycogen content, fat, CHO etc), and with specific sample characteristics (young runners). Additionally, the study protocol and design it is high-quality, and easy to replicate, then I like to recommend the study for future other studies that can help us to carry out other more complex studies.

ABSTRACT

It is well design.

INTRODUCTION

I like to suggest to the authors to include a little more information regarding that pre-exercise H2 ingestion have been reported not at all exercise modalities, intensities, volumes, nor including other outocmes more thatn the “time” in a performance, and population (runners trained).

METHODS

I found the methods reported in a very good quality.

The study protocol figure is very understandable.

Statistical analyses are very clear.

RESULTS

There are very clear described.

DISCUSSION

I like to suggest to the authors to include some future potential other study conditions as I have stated previously (i.e., other populations), where this previous study give us only a high-quality, but very specific condition in which the hydrogen rich water (HRW) ingestion do not add new advantages.

Tables

Are very clear.

Figures

Are very clear.

References

Are very up-dated.

Response: Thank you for your comments and suggestions, particularly relating to the Introduction and Discussion.

Regarding the Introduction, we followed your suggestions, and some of the sentences were rewritten in order to be more specific and explanatory. In addition, we also added the results of some very recent studies that shed more light on the specific exercise intensities, where H2 showed an ergogenic effects. Please, see lines 57–68.

Relating to our approach in the writing of the Discussion, our main goal, particularly in this study, was to investigate whether pre-exercise hydrogen rich water (HRW) intake using our specific hydration strategy (time, dose, chemical properties of HRW) improved running time to exhaustion in very specific population, i.e. trained field and track athletes. To date, we have published six H2 studies in relation to different modes of exercise, intensity level and population. For instance, one study investigated the effect of H2 inhalation on the functional state of post-COVID-19 patients. However, the take home message of the present study was to show coaches that if they decided to use HRW immediately before the race in well trained athletes, HRW does not work well in terms of the expected improvement in running performance across distances from 1500 m to 3000 m. We absolutely agree with your opinion to design future studies that investigate different physiological variables, biomarkers, and population etc. When working with well-trained athletes, there are many practical issues that need to be considered. For example, the research needs to fit around the athletes‘ training programmes and cannot interupt their schedule too much. Nevertheless, our study includes many references to previous studies showing the positive effects of H2 supplementation on fitness, performance, physiological and psychometric responses.

---

## [Decision Letter · Decision Letter 1]

17 Oct 2022

PONE-D-22-15394R1Acute pre-exercise hydrogen rich water intake does not improve running performance at maximal aerobic speed in trained track and field runnersPLOS ONE

Dear Dr. Krejčí,

Thank you for submitting your manuscript to PLOS ONE. After careful consideration, we feel that it has merit but does not fully meet PLOS ONE’s publication criteria as it currently stands. Therefore, we invite you to submit a revised version of the manuscript that addresses the points raised during the review process.==============================

We look forward to receiving your revised manuscript.

Kind regards,

Walid Kamal Abdelbasset, Ph.D.

Academic Editor

PLOS ONE

Journal Requirements:

Reviewers' comments:

Reviewer's Responses to Questions

**Comments to the Author**

1. If the authors have adequately addressed your comments raised in a previous round of review and you feel that this manuscript is now acceptable for publication, you may indicate that here to bypass the “Comments to the Author” section, enter your conflict of interest statement in the “Confidential to Editor” section, and submit your "Accept" recommendation.

Reviewer #1: All comments have been addressed

Reviewer #3: All comments have been addressed

2. Is the manuscript technically sound, and do the data support the conclusions?

Reviewer #1: (No Response)

Reviewer #3: Yes

3. Has the statistical analysis been performed appropriately and rigorously? 

Reviewer #1: (No Response)

Reviewer #3: Yes

4. Have the authors made all data underlying the findings in their manuscript fully available?

Reviewer #1: (No Response)

Reviewer #3: Yes

5. Is the manuscript presented in an intelligible fashion and written in standard English?

Reviewer #1: (No Response)

Reviewer #3: Yes

6. Review Comments to the Author

Reviewer #1: (No Response)

Reviewer #3: The manuscript written in well organized manner, design of the study well constructed, only one comment related to manuscript title as it should contain research type.

7. PLOS authors have the option to publish the peer review history of their article (what does this mean?). If published, this will include your full peer review and any attached files.

Reviewer #1: No

Reviewer #3: No

---

## [Author Response · Author response to Decision Letter 1]

27 Oct 2022

Reviewer #3: The manuscript written in well organized manner, design of the study well constructed, only one comment related to manuscript title as it should contain research type.

Response: Thank you for this suggestion. We added the following words to the title: “A randomized, double-blind, placebo-controlled crossover study”. We looked at the titles of articles in PLOS ONE to see how this type of study is most often writen. We found that a hyphen and a comma are very often used. We therefore rewrote the occurences of this phrase in the abstract and at the beginning of the method section.

---

## [Decision Letter · Decision Letter 2]

6 Dec 2022

Acute pre-exercise hydrogen rich water intake does not improve running performance at maximal aerobic speed in trained track and field runners: A randomized, double-blind, placebo-controlled crossover study

PONE-D-22-15394R2

Dear Dr. Krejčí,

We’re pleased to inform you that your manuscript has been judged scientifically suitable for publication and will be formally accepted for publication once it meets all outstanding technical requirements.

Kind regards,

Walid Kamal Abdelbasset, Ph.D.

Academic Editor

PLOS ONE

Additional Editor Comments (optional):

Reviewers' comments:

Reviewer's Responses to Questions

**Comments to the Author**

1. If the authors have adequately addressed your comments raised in a previous round of review and you feel that this manuscript is now acceptable for publication, you may indicate that here to bypass the “Comments to the Author” section, enter your conflict of interest statement in the “Confidential to Editor” section, and submit your "Accept" recommendation.

Reviewer #3: All comments have been addressed

2. Is the manuscript technically sound, and do the data support the conclusions?

Reviewer #3: Yes

3. Has the statistical analysis been performed appropriately and rigorously? 

Reviewer #3: Yes

4. Have the authors made all data underlying the findings in their manuscript fully available?

Reviewer #3: Yes

5. Is the manuscript presented in an intelligible fashion and written in standard English?

Reviewer #3: Yes

6. Review Comments to the Author

Reviewer #3: All my previous comments had been made. Thanks for your efforts in correcting the manuscript to be processed

7. PLOS authors have the option to publish the peer review history of their article (what does this mean?). If published, this will include your full peer review and any attached files.

Reviewer #3: No

---

## [Editor Report · Acceptance letter]

12 Dec 2022

PONE-D-22-15394R2 

Acute pre-exercise hydrogen rich water intake does not improve running performance at maximal aerobic speed in trained track and field runners: A randomized, double-blind, placebo-controlled crossover study 

Dear Dr. Krejčí:

I'm pleased to inform you that your manuscript has been deemed suitable for publication in PLOS ONE. Congratulations! Your manuscript is now with our production department. 

Kind regards, 

on behalf of

Dr. Walid Kamal Abdelbasset 

Academic Editor

PLOS ONE